# Trajectory of Mini-Batch Momentum: Batch Size Saturation and Convergence in High Dimensions

**Kiwon Lee**
Department of Mathematics and Statistics
McGill University
Montreal, QC H3A 0B9
kiwon.lee@mail.mcgill.ca

**Andrew N. Cheng**
Department of Mathematics and Statistics
McGill University
Montreal, QC H3A 0B9
andrew.cheng@mail.mcgill.ca

**Courtney Paquette**
Department of Mathematics and Statistics
McGill University
Montreal, QC H3A 0B9
courtney.paquette@mcgill.ca

**Elliot Paquette**
Department of Mathematics and Statistics
McGill University
Montreal, QC H3A 0B9
elliot.paquette@mcgill.ca

## Abstract

We analyze the dynamics of large batch stochastic gradient descent with momentum (SGD+M) on the least squares problem when both the number of samples and dimensions are large. In this setting, we show that the dynamics of SGD+M converge to a deterministic discrete Volterra equation as dimension increases, which we analyze. We identify a stability measurement, the implicit conditioning ratio (ICR), which regulates the ability of SGD+M to accelerate the algorithm. When the batch size exceeds this ICR, SGD+M converges linearly at a rate of $\mathcal{O}(1/\sqrt{\kappa})$, matching optimal full-batch momentum (in particular performing as well as a full-batch but with a fraction of the size). For batch sizes smaller than the ICR, in contrast, SGD+M has rates that scale like a multiple of the single batch SGD rate. We give explicit choices for the learning rate and momentum parameter in terms of the Hessian spectra that achieve this performance.

Stochastic learning algorithms are the methods of choice for optimization of high-dimensional problems. Often stochastic learning algorithms incorporate momentum into their stochastic gradients to improve practical performance. Perhaps the simplest, stochastic gradient descent with momentum (SGD+M) adds a fixed multiple of the backward difference of iterates to its stochastic gradient estimator, see Section 1 for details. In the influential work of [33], the authors empirically show augmenting stochastic gradient descent (SGD) with momentum significantly improves training performance of deep neural networks. Despite the wide usage of these stochastic momentum methods in machine learning practice, our understanding of its behaviour is not well–understood.

It has been hypothesized that stochastic-based momentum algorithms improve training because they are employed on a large batch of a data set [15]; thereby emulating the speed-up one sees in full-batch settings. For many learning problems, the "large batch" setting is often paired with high-dimensional problems, meaning there are many samples (and likely also many features to have interesting behavior). While we know of no theoretical work that can fully justify these claims for standard SGD+M, there have been some recent progress [6, 7]. For variations of SGD+M and SGD with Nesterov momentum [24] there has been some success in proving accelerated rates [3, 13, 18]. One reason it has been challenging is that the typical approaches for analyzing SGD+M do not distinguish large and small batch sizes. We address this problem in this paper and we introduce a

36th Conference on Neural Information Processing Systems (NeurIPS 2022).

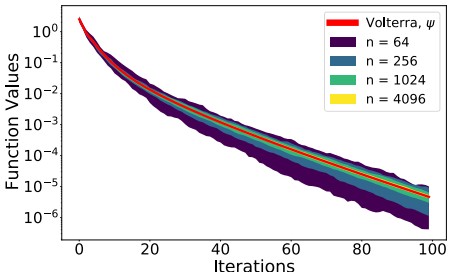

Figure 1: **Concentration of SGD+M on a Gaussian random least squares problem** such that the ratio $d/n$ is fixed to be 2; 30 runs of SGD+M and the 80th percentile confidence intervals recorded (shaded region) for each $n$. The parameters for SGD+M are $\Delta = 0.5, \gamma = 0.4, \zeta = 0.5$, see Section 1.1. The random least squares problem becomes non-random in the large limit and all runs of SGD+M converge to a deterministic function $\psi(t)$ (red) given by our Volterra equation (1).

stability measurement that exactly captures the transition of SGD+M to an accelerated method. We comment that in the high–dimensional, *vanishing batch fraction setting* (the mini-batch size is $o(n)$, where $n$ is the number of samples), there is work proving in various simplified settings that SGD+M produces the same iterates as SGD with a larger learning rate, up to a vanishing error [26].

In this paper, we study the dynamics of mini-batch SGD+M (with constant learning rate) on a least squares problem when the number of samples $n$ and features $d$ are large (see Section 1 for details). We assume the targets are generated using a linear (generative) model. We are motivated by the setting where the mini-batch size $\beta$ is proportionate to the number of samples $n$ and so we define the ratio $\zeta \stackrel{\text{def}}{=} \beta/n$, which we refer to as the *batch fraction*. We provide a non-asymptotic comparison for the behavior of the training loss under SGD+M to a deterministic function $\psi$, whose accuracy improves when the number of samples and features are large while the batch fraction is strictly positive (see Figure 1). This function $\psi$ solves a discrete Volterra equation:

$$\psi(t+1) = F(t+1) + \sum_{k=0}^{t} \psi(k)K(t-k). \quad (1)$$

The *forcing term* $F(t)$ and *kernel* $K(t)$ are explicit functions that depend on the hyperparameters and the full Hessian spectra (see Section 2 and Appendix A). They transparently reveal that the dynamics of SGD+M and of SGD are truly non-equivalent in that there is no mapping of the hyperparameters which leads them to have the same training dynamics. We also note that a similar equation appears in the vanishing batch setting [27], although in that setting it is a Volterra integral equation, which can be recovered from (1) by sending $\zeta \to 0$.

Figure 2: **Convergence rate as batch size changes** on a Gaussian random least squares problem with ratio $r \stackrel{\text{def}}{=} \frac{d}{n}$ varying. Here $\kappa = (1+\sqrt{\frac{1}{r}})^2/(1-\sqrt{\frac{1}{r}})^2$ and $\bar{\kappa} = 1/(1-\sqrt{\frac{1}{r}})^2$. When $\zeta$ goes above a specific value (see Proposition 5), the convergence rate freezes at $1/\sqrt{\kappa}$. Otherwise the rate behaves like $C/\kappa$ with a constant $C = \zeta/(1-\zeta)$ and we get no speed up from SGD+M.

An advantage of the exact loss trajectory is that we give a rigorous definition of the large batch and small batch regimes which reflect a transition in the convergence behavior of SGD+M. To do this we introduce the *condition number* $\kappa$, the *average condition number* $\bar{\kappa}$, and the *implicit conditioning ratio* (ICR) defined as

$$\bar{\kappa} \stackrel{\text{def}}{=} \frac{\frac{1}{n}\sum_{j\in[n]}\sigma_j^2}{\sigma_{\min}^2} < \frac{\sigma_{\max}^2}{\sigma_{\min}^2} \stackrel{\text{def}}{=} \kappa \quad \text{and} \quad \text{ICR} \stackrel{\text{def}}{=} \frac{\bar{\kappa}}{\sqrt{\kappa}}. \quad (2)$$

Here $\sigma_j^2$ are the eigenvalues of the Hessian of the least squares problem with $\sigma_{\max}^2$ and $\sigma_{\min}^2$ the largest and smallest (non-zero) eigenvalues. We refer to the *large batch* regime where $\zeta \geq$ ICR and the *small batch* regime where $\zeta \leq$ ICR. In the large batch regime (2), SGD+M matches the performance of the heavy-ball algorithm: the convergence is linear with rate given by $\mathcal{O}(1/\sqrt{\kappa})$. In the small batch regime, the performance matches that of SGD, i.e. the convergence rate is $\mathcal{O}(\zeta/\bar{\kappa})$. We give matching lower bounds, and we provide momentum and learning rate choices that achieve the claimed performance. In addition we show there is a *saturating batch fraction* (see Figure 2),

after which increasing the batch fraction does not improve the rate. It explicitly occurs when $\zeta =$ ICR. Moreover this saturating batch fraction occurs before full batch, i.e. $\zeta = 1$.

**Related work.** Recent works have established convergence guarantees for SGD+M in both strongly convex and non-strongly convex setting [9, 31], including almost sure convergence [10]. In the work of [25], they used stochastic differential equations (SDEs) to obtain convergence of SGD+M. Specializing to the setting of minimizing quadratics, the authors of [19] demonstrated that the iterates of SGD+M converge linearly (but not in L2) under an exactness assumption.

Determining batch size has been an important issue in determining the convergence rate of SGD and SGD+M. There are instances where (small batch size) SGD+M does not necessarily achieve better performances than small batch size SGD (see [15, 27, 35]). As for mini-batch SGD without momentum, [20] showed that there is a saturating batch size (roughly $\overline{\kappa}/\kappa$) above which increasing the batch size no longer improves the rate. In [8], the authors implemented an adaptive (increasing) batch size schedule and they used it to show linear convergence for SGD. For generalization, [32] empirically showed that for SGD and SGD+M, instead of decaying the learning rate, one can increase the batch size during training to obtain a similar learning curve.

SGD+M has been proven to be useful in practical applications as well, including machine learning. [33] demonstrated that SGD+M shows an empirical advantage in training deep and recurrent neural networks (DNNs and RNNs respectively). Many authors have proposed that learning rate warmup enables us to scale training efficiently to larger batch sizes ([11, 22, 32]).

# 1 Setting

We consider the least squares problem when the number of samples ($n$) and features ($d$) are large:

$$\underset{\boldsymbol{x} \in \mathbb{R}^d}{\arg\min} \left\{ f(\boldsymbol{x}) = \sum_{i=1}^{n} f_i(x) \overset{\text{def}}{=} \frac{1}{2} \sum_{i=1}^{n} (\boldsymbol{a}_i \boldsymbol{x} - b_i)^2 \right\}, \quad \text{with } \boldsymbol{b} \overset{\text{def}}{=} \boldsymbol{A}\widetilde{\boldsymbol{x}} + \boldsymbol{\eta}, \tag{3}$$

where $\boldsymbol{A} \in \mathbb{R}^{n \times d}$ is a data matrix whose $i$-th row is denoted by $\boldsymbol{a}_i \in \mathbb{R}^d$, $\widetilde{\boldsymbol{x}} \in \mathbb{R}^d$ is the signal vector, and $\boldsymbol{\eta} \in \mathbb{R}^n$ is a source of noise. The target $\boldsymbol{b} = \boldsymbol{A}\widetilde{\boldsymbol{x}} + \boldsymbol{\eta}$ comes from a generative model corrupted by noise. We let $\sigma_1^2 \geq \cdots \geq \sigma_n^2 \geq 0$ be the eigenvalues of the matrix $\boldsymbol{A}\boldsymbol{A}^T$ with $\sigma_{\max}^2$ and $\sigma_{\min}^2$ the largest and smallest (nonzero) eigenvalues.

We apply SGD with momentum (SGD+M) with mini-batches to the finite sum, quadratic problem (3). Let $\boldsymbol{x}_0 \in \mathbb{R}^d$ be randomly selected following 1.1 and $\boldsymbol{x}_1$ be generated from SGD without momentum, i.e., $\boldsymbol{x}_1 = \boldsymbol{x}_0 - \gamma \sum_{i \in B_0} \nabla f_i(\boldsymbol{x}_0)$. SGD+M iterates by selecting uniformly at random a subset $B_k \subseteq \{1, 2, \cdots, n\}$ of cardinality $\beta$ and makes the update

$$\boldsymbol{x}_{k+1} = \boldsymbol{x}_k - \gamma \sum_{i \in B_k} \nabla f_i(\boldsymbol{x}_k) + \Delta(\boldsymbol{x}_k - \boldsymbol{x}_{k-1})$$

$$= \boldsymbol{x}_k - \gamma \boldsymbol{A}^T \boldsymbol{P}_k (\boldsymbol{A}\boldsymbol{x}_k - \boldsymbol{b}) + \Delta(\boldsymbol{x}_k - \boldsymbol{x}_{k-1}), \quad \text{where} \quad \boldsymbol{P}_k \overset{\text{def}}{=} \sum_{i \in B_k} \mathbb{e}_i \mathbb{e}_i^T, \tag{4}$$

with $\boldsymbol{P}_k$ a random orthogonal projection matrix and $\mathbb{e}_i$ the $i$-th standard basis vector. Here $\gamma > 0$ is the learning rate parameter, $\Delta$ is the momentum parameter, and the function $f_i$ is the $i$-th element of the sum in (3).

When the stochastic gradient in (4) is replaced with the full-gradient $\nabla f(\boldsymbol{x})$, the resulting algorithm with learning rate and momentum optimally chosen yields the celebrated algorithm, heavy-ball momentum (a.k.a. Polyak momentum) [29]. The optimal learning rate and momentum parameters are explicitly given by

$$\gamma = \frac{4}{(\sqrt{\sigma_{\max}^2} + \sqrt{\sigma_{\min}^2})^2} \quad \text{and} \quad \Delta = \left( \frac{\sqrt{\sigma_{\max}^2} - \sqrt{\sigma_{\min}^2}}{\sqrt{\sigma_{\max}^2} + \sqrt{\sigma_{\min}^2}} \right)^2. \tag{5}$$

It is well-known that heavy-ball is an optimal algorithm on the least squares problem in that it converges linearly at a rate of $\mathcal{O}(1/\sqrt{\kappa})$ (see [28]).

In this paper, we adhere whenever possible to the following notation. We denote vectors in lowercase boldface ($\boldsymbol{x}$) and matrices in upper boldface ($\boldsymbol{A}$). The entries of a vector (or matrix) are denoted by

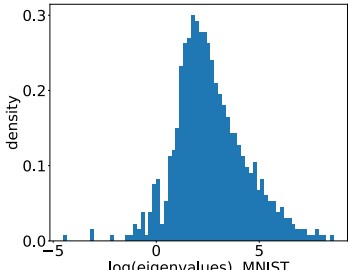 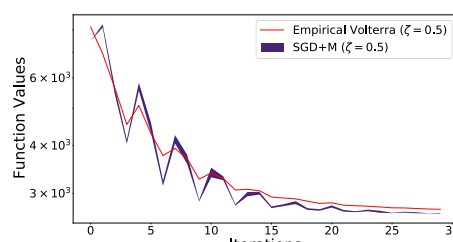

Figure 3: **SGD+M vs. Theory on even/odd MNIST.** MNIST ($60,000 \times 28 \times 28$ images) [16] is reshaped into a single matrix of dimension $60,000 \times 784$ (preconditioned to have centered rows of norm-1), representing 60,000 samples of 10 digits. The target $\boldsymbol{b}$ satisfies $b_i = 0.5$ if the $i^{th}$ sample is an odd digit and $b_i = -0.5$ otherwise. SGD+M was run 10 times with ($\Delta = 0.8, \gamma = 0.001, \zeta = 0.5$) and the empirical Volterra was run once with ($R = 11,000, \tilde{R} = 5300$). The $10^{th}$ to $90^{th}$ percentile interval is displayed for the loss values of 10 runs of SGD+M. While MNIST data set does not satisfy our eigenvalue assumption on the data matrix, the solution to the Volterra equation on MNIST data set captures the dynamics of SGD+M. See App. D for more details.

subscripts. Unless otherwise specified, the norm $\| \cdot \|_2$ is taken to be the standard Euclidean norm if it is applied to a vector and the operator 2-norm if it is applied to a matrix.

## 1.1 Random least squares problem

To perform our analysis we make the following explicit assumptions on the signal $\widetilde{\boldsymbol{x}}$, the noise $\boldsymbol{\eta}$, and the data matrix $\boldsymbol{A}$.

**Assumption 1.1** (Initialization, signal, and noise). *The initial vector $\boldsymbol{x}_0 \in \mathbb{R}^d$ is chosen so that $\boldsymbol{x}_0 - \widetilde{\boldsymbol{x}}$ is independent of the matrix $\boldsymbol{A}$. The noise vector $\boldsymbol{\eta} \in \mathbb{R}^n$ is centered and has i.i.d. entries, independent of $\boldsymbol{A}$. The signal and noise are normalized so that*

$$\mathbb{E}\|\boldsymbol{x}_0 - \widetilde{\boldsymbol{x}}\|_2^2 = R\frac{d}{n}, \quad and \quad \mathbb{E}[\|\boldsymbol{\eta}\|_2^2] = \widetilde{R}.$$

Next we state assumptions on the data matrix $\boldsymbol{A}$ as well as its eigenvalue and eigenvector distribution. Each row $\boldsymbol{a}_i \in \mathbb{R}^{d \times 1}$ is centered and is normalized so that $\mathbb{E}\|\boldsymbol{a}_i\|_2^2 = 1$ for all $i$. We suggest as a central example the *Gaussian random least squares* setup where each entry of $\boldsymbol{A}$ is sampled independently from a standard normal distribution with variance $\frac{1}{d}$.

**Assumption 1.2** (Orthogonal invariance). *Let $\boldsymbol{A}$ be a random $n \times d$ matrix. Suppose these random matrices satisfy a left orthogonal invariance condition: Let $\boldsymbol{O} \in \mathbb{R}^{n \times n}$ be an orthogonal matrix. Then the matrix $\boldsymbol{A}$ is orthogonally left invariant in the sense that*

$$\boldsymbol{O}\boldsymbol{A} \overset{law}{=} \boldsymbol{A}. \tag{6}$$

This assumption implies that the left singular vectors of $A$ are uniformly distributed on the sphere which is the strongest form of eigenvector delocalization; many distributions of random matrices including some sparse ones (such as random regular graph adjacency matrices) are known to have some form of eigenvector delocalization. The classic example of a random matrix which has left orthogonal invariance is the sample covariance matrix, $\boldsymbol{Z}\sqrt{\boldsymbol{\Sigma}}$, for an i.i.d. Gaussian matrix $\boldsymbol{Z}$ and any covariance matrix $\boldsymbol{\Sigma}$. Numerical simulations suggest that (6) can be weakened in that the theory herein can be applied to other ensembles without this orthogonal invariance property. See Figure 3.

## 2 Deterministic Dynamical Equivalent of SGD+M

With these assumptions, we can give an explicit representation of the loss values on a least squares problem at the iterates generated by SGD+M algorithm. We show in this section (see Theorem 1): for any $T > 0$,

$$\sup_{0 \le t \le T} |f(\boldsymbol{x}_t) - \psi(t)| \to 0 \quad \text{in probability,}$$

where $\psi$ solves (1). We begin by discussing the forcing and the noise terms of $\psi(t)$ and their relationship to SGD+M.

**Forcing term: problem instance information.** The forcing term represents the mean (with respect to expectation over the mini-batches) behavior of SGD+M. In fact, the forcing term is the loss $f$ under full-batch gradient descent with momentum $\Delta$ but with learning rate $\gamma\zeta$. For a *small* learning rate $\gamma$, the forcing term $F(t)$ in (1) governs the dynamics of $\psi(t)$.

Let $\boldsymbol{w}_t \stackrel{\text{def}}{=} \boldsymbol{A}\boldsymbol{x}_t - \boldsymbol{b}$ and observe that $1/2\|\boldsymbol{w}_t\|_2^2$ is the loss $f(\boldsymbol{x}_t)$. One way to get the dynamics of $f$ is by deriving a recurrence for $w_{t,j}^2$ (for each $j \in [n]$) which we get from the updates of SGD+M. The recurrence is as followed: define $\tilde{\mathcal{X}}_{t,j} \stackrel{\text{def}}{=} (w_{t,j}^2 \ w_{t-1,j}^2 \ w_{t,j}w_{t-1,j})^T$ and there exists a matrix $\boldsymbol{M}_j$ (see Appendix A.2) so that

$$\tilde{\mathcal{X}}_{t+1,j} = \boldsymbol{M}_j\tilde{\mathcal{X}}_{t,j} + (\text{Error}). \tag{7}$$

The forcing term at iteration $t$ is given by applying a linear recurrence $\boldsymbol{M}_j$ operator $t-1$ times on a vector containing initialization information at each $j \in [n]$ and then summed up for the first coordinate. It is clear that the *maximum* of the eigenvalues of the operator $\boldsymbol{M}_j$ is essential to analyze the convergence behavior of $F(t)$. We denote $\lambda_{1,j} = \Delta$ and $\lambda_{i,j}$, $i = 2, 3$, the eigenvalues of $\boldsymbol{M}_j$ and note that $\Delta \leq \max_{i=2,3}|\lambda_{i,j}|$ (see (39) for an explicit formula of $\lambda_{i,j}$ that depends only on $\gamma$, $\Delta$, and eigenvalues of $\boldsymbol{A}\boldsymbol{A}^T$). Let $\lambda_{2,j}$ be the eigenvalue of $\boldsymbol{M}_j$ with the biggest modulus and let

$$\lambda_{2,\max} \stackrel{\text{def}}{=} \max_j |\lambda_{2,j}|. \tag{8}$$

From this, an equation for the forcing term $F(t)$ can be made explicit in terms of $\lambda_{i,j}$, see Thm. 1 below and Appendix A. We can conclude that $F(t) = \mathcal{O}(\lambda_{2,\max}^t)$.

**Kernel term: noise from the algorithm.** The convolution term in (1) is due to the inherent stochasticity of SGD+M. More specifically, it is given by

$$\gamma^2\zeta(1-\zeta)\sum_{k=0}^{t} H_2(t-k)\psi(k), \quad \text{where} \quad \Omega_j \stackrel{\text{def}}{=} 1 - \gamma\zeta\sigma_j^2 + \Delta,$$

$$\text{and} \quad H_2(t) \stackrel{\text{def}}{=} \frac{1}{n}\sum_{j=1}^{n} \frac{2\sigma_j^4}{\Omega_j^2 - 4\Delta}\left(-\Delta^{t+1} + \frac{1}{2}\lambda_{2,j}^{t+1} + \frac{1}{2}\lambda_{3,j}^{t+1}\right). \tag{9}$$

The presence of $\psi$ (training loss) is due to the fact that the noise generated by the $k$-th stochastic gradient is proportional to $\psi(k)$ (training loss), and the function $H_2(t-k)$ represents the progress of the algorithm in sending this extra noise to $0$. Observe (9) scales quadratically in the learning rate $\gamma$. Hence for *large* learning rates, (9) dominates the decay behaviour of $\psi$. Further details discussed in Section 3.1.

We now state the main result:

**Theorem 1** (Concentration of SGD+M). *Suppose Assumptions 1.1 and 1.2 hold with the learning rate $\gamma < \frac{1+\Delta}{\zeta\sigma_{max}^2}$ and the batch size satisfies $\beta/n = \zeta$ for some $\zeta > 0$. Let the constant $T \in \mathbb{N}$. Then there exists $C > 0$ such that for any $c > 0$, there exists $D > 0$ satisfying*

$$\Pr\left[\sup_{0 \leq t \leq T, t \in \mathbb{N}} |f(\boldsymbol{x}_t) - \psi(t)| > n^{-C}\right] \leq Dn^{-c}, \tag{10}$$

*for sufficiently large $n \in \mathbb{N}$. The function $\psi$ is the solution to the Volterra equation*

$$\psi(t+1) = \underbrace{\frac{R}{2}h_1(t+1) + \frac{\widetilde{R}}{2}h_0(t+1)}_{\text{forcing}} + \underbrace{\sum_{k=0}^{t}\gamma^2\zeta(1-\zeta)H_2(t-k)\psi(k)}_{\text{noise}}, \quad \psi(0) = f(\boldsymbol{x}_0), \tag{11}$$

*where for $k = 0, 1$, $i = 2, 3$, and $j \in [n]$, $\kappa_{i,j} \stackrel{\text{def}}{=} \lambda_{i,j}\Omega_j/(\lambda_{i,j} + \Delta)$, and*

$$h_k(t) = \frac{1}{n}\sum_{j=1}^{n}\frac{2(\sigma_j^2)^k}{\Omega_j^2 - 4\Delta}\left(-\Delta\gamma\zeta(\sigma_j^2)\cdot\Delta^t + \frac{1}{2}(\kappa_{2,j} - \Delta)^2\cdot(\lambda_{2,j})^t + \frac{1}{2}(\kappa_{3,j} - \Delta)^2\cdot(\lambda_{3,j})^t\right).$$

For a more detailed description on $h_0, h_1, H_2$ as well as the proof of Theorem 1 and Corollary 1, see Appendix A. The expression of $\psi$ highlights how the algorithm, learning rate, batch size, momentum, and noise levels interact with each other to produce different dynamics. Note that the learning rate assumption will be necessary for the solution to the Volterra equation to be convergent, see Proposition 2. When $\Delta \to 0$, we obtain the Volterra equation for SGD with mini-batching.

**Corollary 1** (Concentration of SGD, no momentum). *Under the same setting as Theorem 1 and when $\Delta = 0$, the function values $f(\boldsymbol{x}_t)$ converge to $\psi(t)$ as in (10) where now the limit $\psi$ is a solution to the Volterra equation*

$$\psi(t+1) = \frac{R}{2}h_1(t+1) + \frac{\widetilde{R}}{2}h_0(t+1) + \sum_{k=0}^{t}\gamma^2\zeta(1-\zeta)h_2(t-k)\psi(k). \tag{12}$$

*where for $k = 0, 1, 2$,*

$$h_k(t) = \frac{1}{n}\sum_{j=1}^{n}\sigma_j^{2k}(1-\gamma\zeta\sigma_j^2)^{2t}.$$

**Remark.** Note that $H_2(t)$ reduces to $h_2(t)$ in $\Delta = 0$ case. Also when the limit $\zeta \to 0$ and when we scale time by $t/\zeta$, we have that $(1-\gamma\zeta\sigma_j^2)^{2t/\zeta} \to e^{-2\gamma\sigma_j^2 t}$. This coincides with the result from [27, Theorem 1]. Indeed, this shows not only how our dynamics of SGD+M includes the no momentum case (i.e. SGD), but also how the dynamics of SGD+M differ from SGD.

## 3 Convolution Volterra analysis

In this section, we outline how to utilize the Volterra equation (11) to produce a complexity analysis of SGD+M. For additional details and proofs in this section, see Appendix C.

We begin by establishing sufficient conditions for the convergence of the solution to the Volterra equation (11). Our Volterra equation can be seen as the *renewal equation* ([4]). Let us translate (11) into the form of the renewal equation as follows:

$$\psi(t+1) = F(t+1) + (K*\psi)(t), \tag{13}$$

where $(f*g)(t) = \sum_{k=0}^{\infty}f(t-k)g(k)$. Let the *kernel norm* be $\|K\| = \sum_{t=0}^{\infty}K(t)$. By [4, Proposition 7.4], we see that $\|K\| < 1$ is necessary for our solution to the Volterra equation to be convergent. Indeed, we have the following result.

**Proposition 1.** *If the norm $\|K\| < 1$, the algorithm is convergent in that*

$$\psi(\infty) \stackrel{def}{=} \lim_{t\to\infty}\psi(t) = \frac{\frac{\widetilde{R}}{2}(\max\{1-\frac{d}{n},0\})}{1-\|K\|}. \tag{14}$$

Note that the noise factor $\widetilde{R}$ and the matrix dimension ratio $d/n$ appear in the limit. Proposition 1 formulates the limit behaviour of the objective function in both the over-determined and the under-determined case of least squares. When under-determined, the ratio $d/n \geq 1$ and the limiting $\psi(\infty)$ is 0; otherwise the limit loss value is strictly positive. The result (14) only makes sense when the noise term $K$ satisfies $\|K\| < 1$; the next proposition illustrates the conditions on the learning rate and the trace of the eigenvalues of $\boldsymbol{A}\boldsymbol{A}^T$ such that the kernel norm is less than 1.

**Proposition 2** (Convergence threshold). *Under the learning rate condition $\gamma < \frac{1+\Delta}{\zeta\sigma_{max}^2}$ and trace condition $\frac{(1-\zeta)\gamma}{1-\Delta} \cdot \frac{1}{n}\operatorname{tr}(\boldsymbol{A}\boldsymbol{A}^T) < 1$, the kernel norm $\|K\| < 1$, i.e., $\sum_{t=0}^{\infty}K(t) < 1$.*

The *learning rate condition* quantifies an upper bound of good learning rates by the largest eigenvalue of the covariance matrix $\sigma_{\max}^2$, batch size $\zeta$, and the momentum parameter $\Delta$. The *trace condition* illustrates a constraint on the growth of $\sigma_{\max}^2$. Moreover, for a full batch gradient descent model ($\zeta = 1$), the trace condition can be dropped and we get the classical learning rate condition for gradient descent.

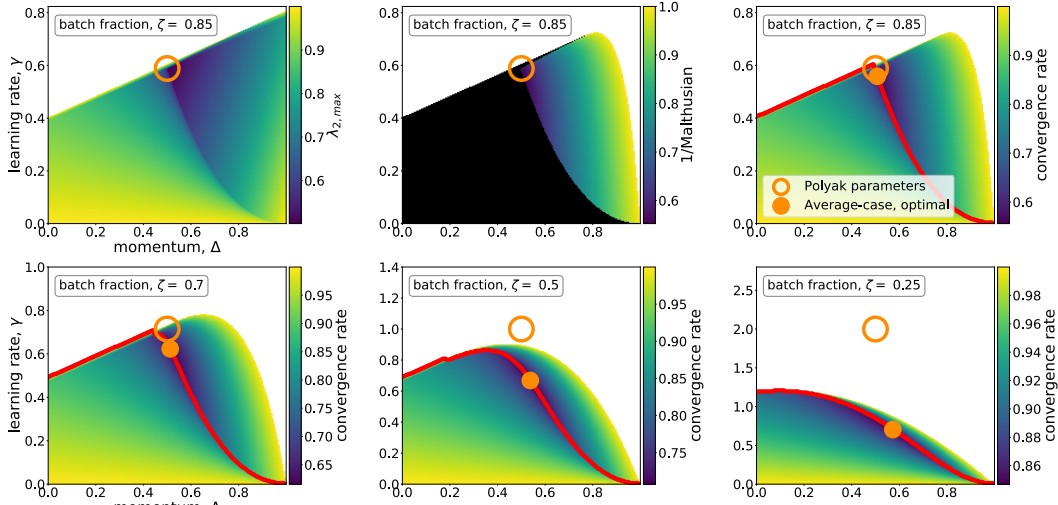

Figure 4: **Different convergence rate regions: problem constrained regime versus algorithmically constrained regime** for Gaussian random least squares problem with ($n = 2000 \times d = 1000$). Plots are functions of momentum ($x$-axis) and learning rate ($y$-axis). Analytic expression for $\lambda_{2,\max}$ (see (8), (39)) – convergence rate of forcing term $F(t)$ – given in (top row, column 1) represents the problem constrained region. (top row, column 2) plots 1/(Malthusian exponent) ((15), for details see Appendix D); black region is where the Malthusian exponent $\Xi$ does not exist. This represents the algorithmically constrained region. Finally, (top row, column 3 and bottom row) plots convergence rate of SGD+M $= \max\{\lambda_{2,\max}, \Xi^{-1}\}$, (see (16)), for various batch fractions. When the Malthusian exponent does not exist (black), $\lambda_{2,\max}$ takes over the convergence rate of SGD+M; otherwise the noise in the algorithm (i.e. Malthusian exponent $\Xi$) dominates. Optimal parameters that maximize $\lambda_{2,\max}$ denoted by Polyak parameters (orange circle, (17)) and the optimal parameters for SGD+M (orange dot); below red line is the problem constrained region; otherwise the algorithmic constrained region. When batch fractions $\zeta = 0.85$ and $\zeta = 0.7$ (top row and bottom row, column 1) (i.e., large batch), the SGD+M convergence rate is the deterministic momentum rate of $1/\sqrt{\kappa}$. As the batch fraction decreases ($\zeta = 0.25$), the convergence rate becomes that of SGD and the optimal parameters of SGD+M and Polyak parameters are quite far from each other. The Malthusian exponent (algorithmically constrained region) starts to control the SGD+M rate as batch fraction $\rightarrow 0$.

### 3.1 The Malthusian exponent and complexity

The rate of convergence of $\psi$ is essentially the worse of two terms – the forcing term $F(t)$ and a discrete time convolution $\sum_{k=0}^{t} \psi(k) K(t - k)$ which depends on the kernel $K$. Intuitively, the forcing term captures the behavior of the expected value of SGD+M and the discrete time convolution captures the slowdown in training due to noise created by the algorithm. Note that $F(t)$ is always a lower bound for $\psi(t)$, but it can be that $\psi(t)$ is exponentially (in $t$) larger than $F(t)$ owing to the convolution term. This occurs when something called the *Malthusian exponent*, denoted $\Xi$, of the convolution Volterra equation exists. The Malthusian exponent $\Xi$ is given as the unique solution to

$$\gamma^2 \zeta (1 - \zeta) \sum_{t=0}^{\infty} \Xi^t H_2(t) = 1, \qquad \text{if the solution exists.} \tag{15}$$

The Malthusian exponent enters into the complexity analysis in the following way:

**Theorem 2** (Asymptotic rates). *The inverse of the Malthusian exponent always satisfies $\Xi^{-1} > \lambda_{2,\max}$ for finite $n$. Moreover, for some $C > 0$, the convergence rate for SGD+M is*

$$\psi(t) - \psi(\infty) \leq C \max\{\lambda_{2,\max}, \Xi^{-1}\}^t \quad \text{and} \quad \lim_{t \to \infty} (\psi(t) - \psi(\infty))^{1/t} = \max\{\lambda_{2,\max}, \Xi^{-1}\}. \tag{16}$$

Thus to understand the rates of convergence, it is necessary to understand the Malthusian exponent as a function of $\gamma$ and $\Delta$.

## 3.2 Two regimes for the Malthusian exponent

On the one hand, the Malthusian exponent $\Xi$ comes from the stochasticity of the algorithm itself. On the other hand, $\lambda_{2,\max}(\gamma, \Delta, \zeta)$ is determined completely by the problem instance information — the eigenspectrum of $AA^T$. (Note we want to emphasize the dependence of $\lambda_{2,\max}$ on learning rate, momentum, and batch fraction.) Let $\sigma_{\max}^2$ and $\sigma_{\min}^2$ denote the maximum and minimum *nonzero* eigenvalues of $AA^T$, respectively. For a fixed batch size, the optimal parameters $(\gamma_\lambda, \Delta_\lambda)$ of $\lambda_{2,\max}$ are

$$\gamma_\lambda = \frac{1}{\zeta}\left(\frac{2}{\sqrt{\sigma_{\max}^2} + \sqrt{\sigma_{\min}^2}}\right)^2 \quad \text{and} \quad \Delta_\lambda = \left(\frac{\sqrt{\sigma_{\max}^2} - \sqrt{\sigma_{\min}^2}}{\sqrt{\sigma_{\max}^2} + \sqrt{\sigma_{\min}^2}}\right)^2. \tag{17}$$

In the full batch setting, i.e. $\zeta = 1$, these optimal parameters $\gamma_\lambda$ and $\Delta_\lambda$ for $\lambda_{2,\max}$ are exactly the Polyak momentum parameters (5). Moreover, in this setting, there is no stochasticity so the Malthusian exponent disappears and the convergence rate (16) is $\lambda_{2,\max}$. We observe from (17) that for all fixed batch sizes, the optimal momentum parameter, $\Delta_\lambda$, is independent of batch size. The only dependence on batch size appears in the learning rate. At first it appears that for small batch fractions, one can take large learning rates, but in that case, the inverse of the Malthusian exponent $\Xi^{-1}$ dominates the convergence rate of SGD+M (16) and you cannot take $\gamma$ and $\Delta$ to be as in (17) (See Figure 4).

We will define two subsets of parameter space, the *problem constrained regime* and the *algorithmically constrained regime* (or stochastically constrained regime). The problem constrained regime is for some tolerance $\varepsilon > 0$

$$\text{problem constrained regime} \quad \stackrel{\text{def}}{=} \{(\gamma, \Delta) : 1 - \sqrt{\Xi} < (1 - \sqrt{\lambda_{2,\max}^{-1}})(1 - \varepsilon)\}. \tag{18}$$

The remainder we call the *algorithmically constrained* regime. To explain the tolerance: for finite $n$, it transpires that we always have $\Xi^{-1} > \lambda_{2,\max}$, but it could be vanishingly close to $\lambda_{2,\max}$ as a function of $n$. Hence we introduce the tolerance to give the correct qualitative behavior in finite $n$.

**Proposition 3.** *If the learning rate $\gamma \le \min(\frac{1+\Delta}{\zeta\sigma_{max}^2}, \frac{(1-\sqrt{\Delta})^2}{\zeta\sigma_{min}^2})$, with the trace condition*

$\frac{8(1-\zeta)\gamma}{1-\Delta} \cdot \frac{1}{n} \operatorname{tr}(A^T A) < 1$, *then $(\gamma, \Delta)$ is in the problem constrained regime with $\varepsilon = 1/2$.*

Therefore by (16), we have that

$$\psi(t) - \psi(\infty) \le D\left(\frac{4\lambda_{2,\max}}{(1 + \sqrt{\lambda_{2,\max}})^2}\right)^t \quad \text{for some } D > 0; \tag{19}$$

we note that the expression in the parenthesis is $1 - \frac{1}{2}(1 - \lambda_{2,\max}) + \mathcal{O}((1 - \lambda_{2,\max})^2)$.

In the problem constrained regime, it is worthwhile to note that the overall convergence rate is the same as full batch momentum with adjusted learning rate, i.e., the batch size does not play an important role as long as we are in the problem constrained regime:

**Proposition 4** (Concentration of SGD + M, full batch)**.** *Suppose $\zeta = 1$ and Assumptions 1.1 and 1.2 hold with the learning rate $\gamma < \frac{1+\Delta}{\sigma_{max}^2}$. If we let $x_t^{full}$ denote the iterates of full-batch gradient descent with momentum (GD+M), then*

$$\sup_{0 \le t \le T} |f(x_t^{full}) - \psi_{full}(t)| \xrightarrow[n \to \infty]{\Pr} 0, \quad \text{where} \quad \psi_{full}(t+1) = \frac{R}{2}h_1(t+1) + \frac{\tilde{R}}{2}h_0(t+1). \tag{20}$$

*The functions $h_1$ and $h_0$ are defined in Theorem 1 with $\zeta = 1$. In particular, let $\gamma_{full}$ denote the learning rate for full batch GD+M, and $\gamma, \zeta < 1$ for the learning rate and batch fraction in SGD+M with corresponding $\psi$ in Theorem 1. Then when $\gamma_{full} = \gamma\zeta$ is satisfied, $\psi$ and $\psi_{full}$ share the same convergence rate in the problem constrained regime.*

## 4 Performance of SGD+M: implicit conditioning ratio (ICR)

Recall from (2) the definition of condition number, average condition number, and the implicit conditioning ratio

$$\bar{\kappa} \stackrel{\text{def}}{=} \frac{\frac{1}{n}\sum_{j\in[n]}\sigma_j^2}{\sigma_{\min}^2} < \frac{\sigma_{\max}^2}{\sigma_{\min}^2} \stackrel{\text{def}}{=} \kappa \quad \text{and} \quad \text{ICR} \stackrel{\text{def}}{=} \frac{\bar{\kappa}}{\sqrt{\kappa}}. \tag{21}$$

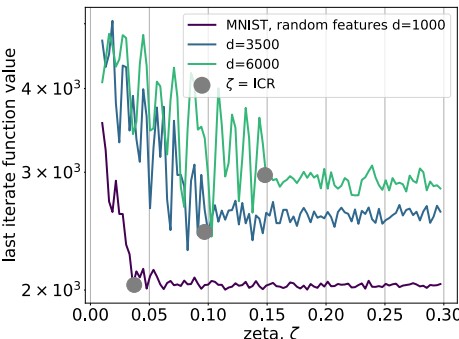

Figure 5: For each value of the batch fraction, $\zeta$, we run SGD+M for 50 iterations on (normalized) MNIST data set using a random features setup with Gaussian weight matrix $\boldsymbol{W} \in \mathbb{R}^{784 \times d}$ (see App. D for details) and targets odd/even. We record the function value of the last iterate. The momentum and learning rate parameters are set to be near-optimal (22). Gray dot is the computed ICR value. At the predicted $\zeta = $ ICR (gray dot), there is a change in the behavior of the last iterate. For $\zeta \leq$ ICR, the value of the last iterate monotonically decreases until it hits the ICR. For $\zeta \geq$ ICR, we see no improvement in the value of the last iterate. This agrees with the theory that the convergence rate does not change.

Moreover recall that we refer to the *large batch* regime where $\zeta \geq$ ICR and the *small batch* regime where $\zeta \leq$ ICR.

We begin by giving a rate guarantee that holds in the problem constrained regime, for a specific choice of $\gamma$ and $\Delta$.

**Proposition 5** (Good momentum parameters). *Suppose the learning rate and momentum satisfy*

$$\gamma = \frac{(1-\sqrt{\Delta})^2}{\zeta \sigma_{\min}^2} \; and \; \Delta = \max\left\{ \left(\frac{1-\frac{\mathcal{C}}{\bar{\kappa}}}{1+\frac{\mathcal{C}}{\bar{\kappa}}}\right), \left(\frac{1-\frac{1}{\sqrt{2\kappa}}}{1+\frac{1}{\sqrt{2\kappa}}}\right) \right\}^2 , \; where \; \mathcal{C} \stackrel{def}{=} \zeta/(8(1-\zeta)). \quad (22)$$

*Then $\lambda_{2,\max} = \Delta$ and for some $C > 0$, the convergence rate for SGD+M is*

$$\psi(t) - \psi(\infty) \leq C \cdot \Delta^t = C \cdot \max\left\{ \left(\frac{1-\frac{\mathcal{C}}{\bar{\kappa}}}{1+\frac{\mathcal{C}}{\bar{\kappa}}}\right), \left(\frac{1-\frac{1}{\sqrt{2\kappa}}}{1+\frac{1}{\sqrt{2\kappa}}}\right) \right\}^{2t} . \quad (23)$$

**Remark 1.** *We note that for all $\Delta$ satisfying $\frac{(1-\sqrt{\Delta})^2}{\zeta \sigma_{\min}^2} \leq \frac{(1+\sqrt{\Delta})^2}{2\zeta \sigma_{\max}^2}$ with the learning rate $\gamma$ as in (22), we have that $\lambda_{2,\max} = \Delta$. By minimizing the $\Delta$ (i.e., by finding the fastest convergence rate), we get the formula for the momentum parameter in (22).*

The exact tradeoff in convergence rates (23) occurs when

$$\frac{\mathcal{C}}{\bar{\kappa}} = \frac{1}{\sqrt{2\kappa}}, \quad \text{or} \quad \zeta = \frac{\frac{8}{\sqrt{2}}\text{ICR}}{1+\frac{8}{\sqrt{2}}\text{ICR}}. \quad (24)$$

As $\zeta \leq 1$, this condition is only nontrivial when ICR $\ll 1$, in which case $\zeta = \frac{8}{\sqrt{2}}$ICR, up to vanishing errors. This is illustrated on the MNIST data set in Figure 5.

**Large batch ($\zeta \geq$ ICR).** In this regime SGD+M's performance matches the performance of the heavy-ball algorithm with the Polyak momentum parameters (up to absolute constants). More specifically with the choices of $\gamma$ and $\Delta$ in Proposition 5, the linear rate of convergence of SGD+M is $1 - \frac{c}{\sqrt{\kappa}}$ for an absolute $c$. Note that $\zeta$ does not appear in the rate, and in particular there is no gain in convergence rate by increasing the batch fraction.

**Small batch ($\zeta \leq$ ICR).** In the small batch regime, the value of $\mathcal{C}$ is relatively small and the first term is dominant in (23), and so the linear rate of convergence of SGD+M is $1 - \frac{c\zeta}{\bar{\kappa}}$ for some absolute constant $c > 0$. In this regime, there is a benefit in increasing the batch fraction, and the rate increases linearly with the fraction. We note that on expanding the choice of constants in small $\zeta$ the choices made in Proposition 5 are

$$\Delta \approx 1 - \frac{\zeta}{8\bar{\kappa}} \quad \text{and} \quad \gamma \approx \frac{\zeta}{256\bar{\kappa}^2 \sigma_{\min}^2}.$$

This rate can also be achieved by taking $\Delta = 0$, i.e. mini-batch SGD with no momentum. Moreover, it is not possible to beat this by using momentum; we show the following lower bound:

**Proposition 6.** *If* $\zeta \leq \min\{\frac{1}{2}, ICR\}$ *then there is an absolute constant* $C > 0$ *so that for convergent* $(\gamma, \Delta)$ *(those satisfying Proposition 2),* $\sqrt{\lambda_{2,\max}} \geq 1 - \frac{C\zeta}{\kappa}$.

This is a lower bound on the rate of convergence by Theorem 2.

A parallel argument of Proposition 5 holds for SGD without momentum.

**Proposition 7** (Good learning rate for SGD without momentum). *Suppose the learning rate* $\gamma$ *satisfies*

$$\gamma = \max\left\{ \frac{1}{\zeta\sigma_{\max}^2}, \frac{1}{8(1-\zeta) \cdot \frac{1}{n}\operatorname{tr}(\boldsymbol{A}^T\boldsymbol{A})} \right\}. \tag{25}$$

*Then* $\lambda_{2,\max} = (1 - \gamma\zeta\sigma_{\min})^2$ *and for some* $C > 0$, *the convergence rate for SGD without momentum is*

$$\psi(t) - \psi(\infty) \leq C \cdot \lambda_{2,\max}^t = C \cdot \max\left\{ 1 - \frac{\mathcal{C}}{\bar{\kappa}}, 1 - \frac{1}{\kappa} \right\}^{2t}, \tag{26}$$

*where* $\mathcal{C} \stackrel{\text{def}}{=} \zeta/(8(1-\zeta))$.

For details on the proof of Proposition 7, see Appendix C. The exact tradeoff in convergence rates occurs when

$$\frac{1}{\zeta\sigma_{\max}^2} = \frac{1}{8(1-\zeta) \cdot \frac{1}{n}\operatorname{tr}(\boldsymbol{A}^T\boldsymbol{A})}, \quad \text{or} \quad \zeta = \frac{8 \times \widehat{ICR}}{1 + 8 \times \widehat{ICR}}, \quad \text{where} \quad \widehat{ICR} \stackrel{\text{def}}{=} \frac{\bar{\kappa}}{\kappa}. \tag{27}$$

As $\zeta \leq 1$, this condition is only nontrivial when $\widehat{ICR} \ll 1$, in which case $\zeta = 8 \times \widehat{ICR}$, up to vanishing errors. When we are in the large batch setting ($\zeta \gtrsim \widehat{ICR}$), the linear rate of convergence of SGD is $1 - \frac{c\zeta}{\bar{\kappa}}$ for some absolute constant $c > 0$.

On the other hand, when in the small batch regime, i.e., $\zeta \lesssim \widehat{ICR}$, the convergence rate is fixed by $1 - \frac{1}{\kappa}$. Note that $\zeta$ does not appear in the convergence rate, so there is no loss in convergencec rate by decreasing the batch fraction.

As a result, SGD converges more slowly in the large batch setting than in the small batch setting (see [14]).

## 5  Conclusion and future work

We have shown that the SGD+M method on a least squares problem demonstrates deterministic behavior in the large $n$ and $d$ limit. We described the dynamics of this algorithm through a discrete Volterra equation and for a fixed batch fraction. Moreover we characterized a dichotomy of convergence regimes depending on the learning rate and momentum parameters. Furthermore, we proved that SGD+M shows a distinguishable improvement over SGD in the large batch regime and we provided parameters which achieve acceleration. Our theory is also supported by numerical experiments on the isotropic features model and MNIST data set (see Appendix D for details).

While our analysis focuses on SGD+M algorithm applied to the least squares problems with orthogonal invariant data matrix, Figure 3 suggests that the Volterra prediction might hold in even greater generality. Removing these conditions, we leave as future work. Another direction of future work consists in finding the deterministic dynamics for generalization errors.

## Acknowledgments and Disclosure of Funding

C. Paquette's research was supported by CIFAR AI Chair, MILA, a Discovery Grant from the Natural Science and Engineering Council (NSERC), and the FRQNT New University Researcher's Start-up Program. Research by E. Paquette was supported by a Discovery Grant from the Natural Science and Engineering Council (NSERC). Additional revenues related to this work: C. Paquette has part-time employment at Google Research, Brain Team, Montreal, QC.

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
