# OpenReview forum: "Trajectory of Mini-Batch Momentum: Batch Size Saturation and Convergence in High Dimensions"
_NeurIPS.cc/2022/Conference — NeurIPS 2022 Accept_

### Official Review · Reviewer_ARN4 · 2022-06-15

**Rating:** 8
**Confidence:** 3
**Soundness:** 3 good
**Presentation:** 2 fair
**Contribution:** 3 good

**Summary:**

Analysis and characterization of two regimes for SGD on quadratic problems. Large batch setting where accelerated rates occur and small batch setting where non-accelerated rates occur.

**Questions:**

It's confusing to use \kappa as a constant and capital kappa for kernel, the symbols are almost the same, perhaps change the notation for the kernel to just K? a lot of papers use that.

The beginning of section two doesn't make much sense without reading the appendix, it leaves too much out.

Are the eigenvalues of H $\sigma$ or $\sigma^{2}$? Using $\sigma^{2}$ is a little odd.

**Limitations:**

A theory paper is always limited by its assumptions. The assumptions are very very strong here, almost the strongest possible given the setting. However, the results are correspondingly concrete and specific, so strong assumptions are ok.

**Strengths And Weaknesses:**

I generally like the approach this paper takes and the theoretical outcomes. The separation of the convergence rates into two regimes makes a lot of sense and matches my personal intuition about the behavior occurring in practice. Although momentum helps stochastic optimization in many practical settings, in my experience it doesn't appear to provide "acceleration", but rather robustness and generalization advantages, and I think the behavior in practice is very much what the theory in this paper describes in the small batch setting. I don't know of any papers using large-batch training where acceleration-like improvements are observed when using momentum, I wonder if the authors are aware of any?
I'm not familiar with discrete Volterra equations, as I expect would be the case for most readers. I'm still able to largely follow the arguments presented. The results are surprisingly precise regarding the $\zeta$ and its relation to the spectrum. Some further plots showing the behavior in and around the cross-over region would be nice.

The use of text-wrapped figures here doesn't make sense to me, given that in both cases two figures are stacked, they could easily be instead stacked side-by-side and floated at the top of the pages without losing any space, I would suggest the authors do that for the camera ready. It's more readable.

---

> ### Author Response · Authors · 2022-08-02
> **Response to Reviewer ARN4**
>
> We thank the reviewer for the careful reading of our paper and the reviewer's appreciation for our contributions and general approach to analyzing SGD+M.
>
> We have implemented the following in the **newly uploaded version of the Supplementary Material and main text** as suggested by the reviewer. We encourage the reviewer to look at these in the new versions.
>
> * Figure wrapping changes.
> * Added experiments showing behavior near the **[crossover batch fraction point for simulated data and MNIST in Appendix D of Supplementary Materials (see here as well)](https://anonymous.4open.science/r/predicted_ICR-C2F8/figure7.pdf)**
> * Removed $\mathcal{K}$ to represent the kernel
> * Rewrote beginning of Section 2 to help with clarification. We would like more feedback from the reviewer on this.
>
> **General comments**
>
> *“I don't know of any papers using large-batch training where acceleration-like improvements are observed when using momentum, I wonder if the authors are aware of any?”*
>
> There are two concurrent works of ours [1,2] that have shown some degree of acceleration. These were released after and right before (~1.5 months) our submission to NeurIPS (thus making them concurrent works according to NeurIPS).
>
> * In [1], the authors showed that SGD+M (and other related stochastic accelerated algorithms such as stochastic Nesterov’s accelerated method) achieve the $O(1/\sqrt{\gamma \sigma^2_{\min}} )$ rate where $\gamma$ is the learning rate, but require $\gamma$ small so one may not see the acceleration.
> * In [2], the authors, using a finer analysis of products of random matrices, show that mini-batch SGD+M achieves the desired $1/\sqrt{\kappa}$ rate provided that the mini-batch is sufficiently large; a similar ideas as us. To achieve this acceleration, the authors require that their batch fraction be an order $\kappa$ bigger than the ICR we propose. This extra condition number can be quite large which in turn means they need huge batch sizes to achieve acceleration. Moreover in their experiments they show that our batch fraction is sufficient. This said, they work in a more general setting. **Moreover in [our new figure (see here)](https://anonymous.4open.science/r/predicted_ICR-C2F8/figure7.pdf), you can see that when the batch fraction equals the ICR, it *exactly* finds the transition point.**
>
> [1] Can, B. and Gurbuzbalaban, M.  *Entropic Risk-Averse Generalized Momentum Methods*, arxiv preprint: 2204.11292
> [2] Bollapragada, R., Chen, T., and Ward, R. *On the fast convergence of minibatch heavy ball momentum*, arxiv preprint: 2206.07553
>
> *Some further plots showing the behavior in and around the cross-over region would be nice.*
>
> We have added in the Supplementary Materials, Appendix D a new experiment showing the behavior near the ICR region for simulated data and on MNIST **[(see Figure 7 or the link here)](https://anonymous.4open.science/r/predicted_ICR-C2F8/figure7.pdf)**. The results show that the ICR *exactly* predicts the crossover behavior even on real data such as MNIST.
>
> *Removing text-wrapping of figures.*
>
> We implemented the reviewer’s suggestion in the new revision of the paper for some of the figures. Unfortunately for one figure we could not remove the text-wrapping without causing us spacing issues. If the reviewer has another suggestion to fix this, we would gladly implement it.
>
> **Response to Questions**
>
> *1. It's confusing to use \kappa as a constant and capital kappa for kernel, the symbols are almost the same, perhaps change the notation for the kernel to just K? a lot of papers use that.*
>
> We thank the reviewer for their feedback and we have made the revision in the new version of the paper.
>
> *2. The beginning of section two doesn't make much sense without reading the appendix, it leaves too much out.*
>
> We have rewritten the beginning of Section 2 and added some of the equations from the Appendix to clarify the forcing term in the new version of the paper.
>
> *3. Are the eigenvalues of H σ or σ2? Using σ2 is a little odd.*
>
> The $\sigma^2$ are the eigenvalues of $AA^T$; we used this notation so that $\sigma$ would represent a singular value of $A$. As far as we are aware, there is no $H$ in the paper. Could the reviewer clarify what $H$ they are referring to?

---

### Official Review · Reviewer_qtmY · 2022-07-05

**Rating:** 6
**Confidence:** 3
**Soundness:** 3 good
**Presentation:** 2 fair
**Contribution:** 3 good

**Summary:**

In this paper, the authors analyzed the dynamics of stochastic gradient descent with momentum based on the linear regression (least-squares) problems. They used the deterministic discrete Volterra equation to approximate the dynamics of stochastic gradient descent with momentum. They provided proof of convergence of the iterates of stochastic gradient descent with momentum to the discrete Volterra equation in Theorem 1. They proposed the Implicit conditioning ratio (ICR) and claimed that there are two regimes. For large batch sizes, the SGD+M performance matches the performance of the heavy-ball algorithm with Polyak momentum parameters, while there is no gain in convergence rate by increasing the batch fraction. For small batch sizes, there is still a benefit in increasing the batch fraction.

**Questions:**

1) In Assumption 1.1, there is an assumption about the initialization x_0, which is not normal for the problems. It is hard to check this assumption since the signal \tilde{x} is not known in practice.
2) In the experiment part, Figures 4 and 6 are based on Gaussian random least squares problems. They are based on calculating the analytic expression for convergence rate. Is it possible to plot the figure based on an empirical experiment?
3) In section 4, the authors mentioned the two batch size regimes. Providing experiments comparing these two regimes will be great to validate this claim.
4) Figure 4 and 6 are based on least-squares problems, are there similar figures for MNIST experiments?

**Limitations:**

Yes

**Strengths And Weaknesses:**

Strengths:
1) The paper has solid mathematical proof and contributions.
2) The figures in the paper are well-presented.
3) The idea of the paper has been written in a clear and reader-friendly way.

Weaknesses:
1) The organization of the paper is not reader-friendly. The figures of experiments are located in different places on paper and the setting of the experiments is in the supplementary documents. It's quite hard to follow when reading the paper.
2) There are several functions in the paper, such as h_1, h_0, H_2, which are defined in the supplementary documents. But in the main text, the authors use the conclusion from the function that F(t)-\bigO(\lambda _{2,max}^t). This may confuse readers.
3) Since there are several constants in the paper, it may be better to have a table for the definition of constants.
4) There is not enough related work discussed in the related work section.

---

> ### Author Response · Authors · 2022-08-02
> **Response to Reviewer qtmY**
>
> We thank the reviewer for their suggestions about further experiments,  which we added to the newly uploaded Supplementary Materials. In particular, we did the following:
>
> * Reran Figure 4 – Convergence rate for various momentum and stepsizes on MNIST data. The figure is found in Supplementary Materials, Appendix D (Figure 6) **[see here](https://anonymous.4open.science/r/heat_maps-B391/figure6.pdf)**.
> * Added Figure 7 -- Convergence behavior near batch fraction equal to ICR on simulated data and MNIST **[(see here)](https://anonymous.4open.science/r/predicted_ICR-C2F8/figure7.pdf)**. The ICR *exactly* predicted the transition.
> * Added the definitions of  $h_1, h_0, H_2$ into the main text.
>
> We appreciate that the reviewer likes the contributions of the paper. We clarify below the reviewer’s questions and concerns.
>
> **Response to Weaknesses**
>
> *1. The organization of the paper is not reader-friendly. The figures of experiments are located in different places on paper and the setting of the experiments is in the supplementary documents. It's quite hard to follow when reading the paper.*
>
> We fixed the figure placement in the revision and hopefully the placement is better in the updated version of the paper which we encourage you to look at. Unfortunately due to space limitations, we can not fit the experimental set-up into the main 9 pages without removing some of the important theoretical results.
>
> *2. There are several functions in the paper, such as $h_1, h_0, H_2$, which are defined in the supplementary documents. But in the main text, the authors use the conclusion from the function that $F(t)-O(\lambda _{2,max}^t$). This may confuse readers.*
>
> We have added the definitions of  $h_1, h_0, H_2$ into the main text.
>
> *3. Since there are several constants in the paper, it may be better to have a table for the definition of constants.*
>
> Due to space limitations, we do not have the room to place a table into the main 9 pages.
>
> *4. There is not enough related work discussed in the related work section.*
>
> We will add the following references if accepted; we were limited on space but will add if we are given the extra 1 page to the main text. (If the reviewer has more suggestions, we would be grateful.)
> * In Gitman, et al [2019], the authors use an analysis of SGD+M which breaks down the rate in two: a deterministic part and a part governed by the noise. They derive an exact formula in the most general quadratic setting, but do not analyze the rates to show if SGD+M achieves $O(1/\sqrt{\kappa})$.
> * Other works have shown that worst-case SGD+M does not achieve acceleration (see e.g., Kidambi et al. [2018], Sebbouh et al. [2020], Zhang et al. [2019], Liu et al [2020]).
> * The lack of general convergence guarantees showing acceleration for existing momentum schemes, such as heavy-ball, in the stochastic setting, has led many authors to design alternative acceleration schemes [Allen-Zhu, 2017, Ghadimi and Lan, 2012, 2013a, Kidambi et al., 2018, Kulunchakov and Mairal, 2019, Liu and Belkin, 2020].
>
> **Response to Questions**
>
> *1. In Assumption 1.1, there is an assumption about the initialization x_0, which is not normal for the problems. It is hard to check this assumption since the signal \tilde{x} is not known in practice.*
>
> The condition in Assumption 1.1 says that $||x_0-\tilde{x}||^2$ is bounded, which is a common assumption in theory. The $d/n$ appearing was for convenience in the analysis. The assumption is simply that $\||x_0-\tilde{x}\||^2$ is bounded (or the distance to optimality) does not grow in dimension, that is as $n \to \infty$ or $d \to \infty$. We do not want this quantity to grow in $n, d$ because we are varying the dimensions $d$ and $n$, and so we need to ensure that as $d$ and $n$ grow the distance to optimality $\||x_0-\tilde{x}\||^2$ is constant so that we don’t start further away from the optimum (e.g., if $x_0 \sim N(0,I_d)$ then as $E[||x_0||^2]$ grows like d).
>
> Note one does not actually need to know $R$ and $\tilde{R}$ in order to compute $\psi$ (see Reviewer n1XZ, Question 3).
>
> *2. In the experiment part, Figures 4 and 6 are based on Gaussian random least squares problems. They are based on calculating the analytic expression for convergence rate. Is it possible to plot the figure based on an empirical experiment?*
>
> We have added in Appendix D, Supplementary Materials (see new revision of paper) a **[new figure, Figure 6 (see here)](https://anonymous.4open.science/r/heat_maps-B391/figure6.pdf)**, which is Figure 4 but on the MNIST dataset.
>
> *3. In section 4, the authors mentioned the two batch size regimes. Providing experiments comparing these two regimes will be great to validate this claim.*
>
> See **[Figure 7 in Appendix D of Supplementary Material or here](https://anonymous.4open.science/r/predicted_ICR-C2F8/figure7.pdf)** showing behavior near the ICR. The batch fraction equal to the ICR *exactly* predicted this crossover behavior.

---

> > ### Comment · Reviewer_qtmY · 2022-08-09
> > **Thanks for the response**
> >
> > Thanks for addressing the questions and weaknesses in my comment. I have raised the score according to the responses.

---

### Official Review · Reviewer_hZQR · 2022-07-11

**Rating:** 6
**Confidence:** 3
**Soundness:** 3 good
**Presentation:** 3 good
**Contribution:** 3 good

**Summary:**

This paper studies large batch stochastic gradient descent with momentum, where the setting is specialized to the least square problem with large sample size and dimension. It is shown that the dynamics of SGD with momentum will converge to deterministic dynamics described by the discrete Volterra equation, as the sample size increases. Based on this, a quantity called implicit conditioning ratio is identified to characterize when SGD with momentum can achieve acceleration. This further yields suitable choices for the learning rate and momentum parameter.

**Questions:**

Some comments and questions are as follows:

1. The paper title suggests that the dimension $d$ is high, but this is not clear from the results. For example, in Theorem 1, is it true that the result holds for a fixed $d$ as $n$ goes to infinity? This also concerns the paragraph from line 220 to 225. When $d$ is fixed, it is an over-determined linear regression for large $n$. The same problem exists in the statement of Proposition 4. If $d$ is fixed and $n$ is large, then basically each SGD step corresponds to an over-determined linear regression?  I'd like the authors to clarify the relationship between $n$ and $d$.

2. In Theorem 1, it is said that Eq.(12) holds for sufficiently large $n$. What is the threshold for being 'sufficiently large'? Also, the definition/properties of the function $H$ in Eq.(13) should be specified in the theorem statement, otherwise it is not clear what is its dependence on the problem parameters.

3. What is $\mathcal{K}$ in Eq.(15) when translating Eq.(13) into the form of Eq.(15)?

4. Theorem 1 characterizes the closeness between the SGD+M iterates and the discrete Volterra equation iterates, but only for finite $T$ and the error does not decrease with $T$. This suggests that there is a gap between the convergence of SGD+M and that of $\psi$. Thus it is misleading to make claim about the convergence of SGD+M based on the convergence of $\psi$, as in Theorem 2. I'd like the authors to discuss how to resolve this gap, otherwise the claims about the convergence of SGD+M is problematic.

**Limitations:**

The authors have adequately addressed the limitations and potential negative societal impact of their work.

**Strengths And Weaknesses:**

This paper is clearly written and easy to follow. The problem setting and assumptions are clarified, and the discussion on the results yields interesting observations about the behavior of SGD with momentum. However, there is still a gap between the actual convergence of SGD with momentum and the presented equivalence between SGD with momentum and the discrete Volterra equation, which makes some claims a bit misleading. See details below.

---

> ### Author Response · Authors · 2022-08-02
> **Response to Reviewer hZQR**
>
> We would like to clarify some concerns of the reviewer. We hope that the reviewer will reconsider their rating and carefully read our response on how to interpret Theorem 1 and, for instance, obtain convergence rates.
>
> **Response to Weaknesses**
>
> *“…gap between the actual convergence of SGD+M and the presented equivalence between SGD+M and the discrete Volterra equation, which makes some claims a bit misleading…Addressing Theorem 1 characterizes the closeness between the SGD+M iterates and the discrete Volterra equation iterates, but only for finite T and the error does not decrease with T...Thus it is misleading to make claim about the convergence of SGD+M based on the convergence of ψ, as in Theorem 2.”*
>
> The subject of this article is high--dimensional analysis: we assume that the number of samples $n$ is large (and are interested in cases when both $n$ and $d$ are large), and we look for ways in which the problem simplifies over low dimensions.  As it is typically the case in machine learning problems that these are large (in the hundreds is already enough for good numerical agreement), this assumption is usually desirable.
>
> **The Volterra model that we derive is the explicit high-dimensional ($n$ is large) equivalent of the minibatch suboptimality.**
>
> **In our theorem formulation, the quality of the agreement is high when $n$ is large,** and as you observe it does not improve with iteration count.  Formulating a statement that holds on an **infinite time horizon is possible**, and could be an interesting future direction.
>
> But the convergence can not already be determined from our theorem, although the formulation is different from non-high-dimensional optimization.  You should rather use the following approach.
>
> To reach $\varepsilon$-suboptimality in the minibatch SGD+M, you would first use the Volterra equation to derive the number of iterates T required to reach this suboptimality.  Our theorem then tells you a dimensionality in which you should work (although we do not quantify it), and then it says with overwhelming probability the random algorithm reaches suboptimality $2\varepsilon$ in $T$ steps.
>
> So as a consequence, our theorem shows that the trajectory of the minibatch training loss is contained in a tube around a particular path, which implies neighborhood convergence.  Of course in the non-interpolation regime, with a stochastic algorithm and without iterate averaging and with constant step size, this is the best you can hope for anyhow.  Studying those alterations is also possibly interesting future work.
>
> We would like to emphasize that while our convergence is 'neighborhood' type, it is substantially sharper than any other convergence result you will find in the literature in high dimensions.  The Volterra equation is **actually the high--dimensional limit**.  This means that to reach a given $\varepsilon$-suboptimality, there is a number $T_{\varepsilon}$ (determined by the Volterra equation) so that mini batch-SGD+M requires *exactly* $T_{\varepsilon}$ steps to reach this suboptimality with overwhelming probability, for all dimensions sufficiently large.
>
> **Response to Questions**
>
> *1. The paper title suggests that the dimension d is high, but this is not clear from the results. In Theorem 1, is it true that the result holds for a fixed d as n goes to infinity? If d is fixed and n is large, then basically each SGD step corresponds to an over-determined linear regression? I'd like the authors to clarify the relationship between n and d.*
>
> The results hold for any $n$ and $d$ provided $n$ is large (and, as suggested by the title of the paper, we are interested when both $n$ and $d$ are large as the “over-parameterized, high-dimensional” setting is common in machine learning settings) . This means that one can have $d \ge n$ or $n \ge d$; both the over- and under- determined settings are represented by all the theorems and propositions. The only place large $d$ enters is the concentration related to the signal R (Assumption 1.1), but if we condition on $x_0$, $\tilde{x}$ and set $R = ||x_0-\tilde{x}||^2$, there is nothing else regulated by $d$. Thus one can take $d$ fixed and $n \to \infty$ and all propositions and theorems hold.
>
> *2. In Theorem 1, it is said that Eq.(12) holds for sufficiently large n. What is the threshold for being 'sufficiently large'?Definition/properties of the function H in Eq.(13) should be specified in the theorem statement*
>
> The “sufficiently large n” is to suppress a constant that appears in Thm 1. For exactly how large an $n$, it should satisfy Line 549 in Lem. 1 in the Appendix.
>
> Added the definitions of $h_1, h_0,$ and $H_2$ (and their dependencies on $\gamma, \Delta, \zeta$, $A A^T$) to the statement of Theorem 1 in the new version of the paper.
>
> *3. What is K in Eq.(15) when translating Eq.(13) into the form of Eq.(15)?*
>
> K in Eq. (15) is $K(t) = \gamma^2 \zeta (1-\zeta) H_2(t)$ where $H_2(t)$ is defined in Eq. 9, see Appendix A for details.

---

> > ### Comment · Reviewer_hZQR · 2022-08-06
> > **Thanks for the clarifications**
> >
> > First I thank the authors for the explanations. I have read other reviewers' reviews and the corresponding rebuttal.
> >
> > I find one thing confusing: The authors mentioned in the response that 'one can take $d$ fixed and $n \to \infty$ and all propositions and theorems hold'. Let's say $d$ is fixed, then as $n \to \infty$, Assumption 1 tells us that the initialization $x_0$ is arbitrarily warm as $E[||x_0 - \tilde x||_2^2] = \frac{Rd}{n} \to 0$. However, Proposition 1 instead tells us that $\psi(\infty) = \frac{\frac{\widetilde R}{2} (\max(1-\frac{d}{n}, 0))}{1-||K||} \to \frac{\widetilde R}{2(1-||K||)}$. Is this a contradiction? When $x_0$ is very close to $\tilde x$, then basically $f(x_0) = \frac{1}{2} ||\eta||_2^2 = \frac{\widetilde R}{2}$ So $||K||=0$ in this case? In other words, the case of $d$ being fixed degenerates due to Assumption 1, which requires an arbitrarily warm start.
> >
> >
> > Also, it seems to me that the analysis is limited to the case of linear model where $b = Ax + \eta$. Otherwise it is not clear to me how the high-dimensional effect kicks in, which crucially relies on the property of matrix $A$ in the current setting. If this is true, I think the limitation should be stated clearly somewhere early in the paper.

---

> > > ### Author Response · Authors · 2022-08-06
> > > **Clarification**
> > >
> > > **Limitation of $b= A\tilde{x} + \eta$**
> > >
> > > In the current set up of the paper, we do assume that $b = A\tilde{x} + \eta$; a generative model, that is, the targets come from a true signal plus noise. The set-up is widely used (see [1,2] to name a few).  We *explicitly* state this in **Eq 3** in the set-up section of the paper. Moreover as mentioned below, we can eliminate this assumption.
> > >
> > > [1] Mei, S. and Montanari, A. *The generalization error of random features regression: Precise asymptotics and double descent curve*, 2019
> > >
> > > [2] Adlam, B. and Pennington, J. *The Neural Tangent Kernel in High Dimensions: Triple Descent and a Multi-Scale Theory of Generalization*, 2020
> > >
> > > **Response to $d$ fixed and $n \to \infty$**
> > >
> > > First in order to take $d$ fixed and $n \to \infty$, as mentioned in our rebuttal, one would have to change Assumption 1.1 because, as you mentioned, this would imply that $E[\||x_0-\tilde{x}\||^2_2] \to 0$. However you can reinterpret what the $R$ and $\tilde{R}$ mean in Assumption 1.1. A better way (which we did not state in the paper, but can do so), is to view the whole forcing term $F(t) =  \frac{R}{2} h_1(t) + \frac{\tilde{R}}{2} h_0(t)$ as the loss function $f$ applied to full-batch gradient descent with momentum but with learning rate $\gamma * \zeta$. We note the forcing term is the only place where $x_0, \tilde{x}$, and $\eta$ are explicitly used. In this way, you can start with any $x_0$ and $b$ under some mild assumptions (i.e., $\||x_0\||^2 \le C$ and $\||b\||^2 \le C$ independent of $n$ and $d$). Then the dynamics would be
> > >
> > > $$ \psi(t+1) = f(x_k^{\text{full batch gd+M}}) + \gamma^2 \zeta (1-\zeta) \sum_{k=0}^t H_2(t-k) \psi(k), \text{where full batch gd+m is with stepsize $\gamma \zeta$.}$$
> > > If the reviewer prefers this interpretation, we can add it to the paper. This has the nice view of showing exactly that loss under SGD+M is loss under GD+M plus a noise term which is generated by the stochasticity in algorithm and we give explicit expressions for the noise.
> > >
> > > One nice reason to state with $R$ and $\tilde{R}$ is that $R/\tilde{R}$ represents the signal-to-noise ratio. Often many high-dimensional asymptotic results use this quantity.
> > >
> > > We also want to point out that most ML architectures are in the regime where $d$ is proportional to $n$. [3]
> > >
> > > [3] Martin, C. and Mahoney, M. *Implicit Self-Regularization in Deep Neural Networks: Evidence from Random Matrix Theory
> > > and Implications for Learning*, 2021
> > >
> > > **Confusion about $x_0$ close to $\tilde{x}$ and kernel** ($||K||$ not necessarily $0$ and no contradiction).
> > >
> > > We are not assuming that the step size $\gamma \to 0$. Therefore because we are running a stochastic algorithm, we always move away from $x_0$ in one step provided the step size is not $0$. This is the noise created by running any stochastic algorithm such as SGD+M (i.e., you always take a step away from the optimum even if you are at the optimum since the *stochastic gradient* is not $0$). The $1-\||K\||$ is capturing the cumulative effect of this noise as $k \to \infty$.
> > >
> > > We note that if the step size goes to $0$, then $\||K\|| \to 0$ as $k \to \infty$ (see Eq. 11). You expect this because it is known that SGD with step size $\to 0$ converges to the noise in $b$. We agree with the reviewer, in this setting with $\gamma \to 0$, that the loss value does not change much from initialization.
> > >
> > > On the other hand, if the step size does not go to $0$, there is persistent noise from the algorithm (i.e., you always take a step away from the optimum even if you are at the optimum). The effect is that $0< \||K\|| < 1$. The value at $\infty$ reflects the cumulative noise of always having to take a step because $\gamma > 0$. Our model captures both when stepsize goes 0 and when stepsize is bounded away from $0$.
> > >
> > > *Is this what the reviewer is concerned about? Could the reviewer elaborate more on the "contradiction" part?*
> > >
> > > There is also **no warm start needed** because $R$ is arbitrary. For instance, fix any dimension (say $n> d$) and chose an $x_0$ and assume that $b = A\tilde{x} + \eta$. Then there exists an $R$ such that $\||x_0-\tilde{x}\||^2 = R d/ n$. We then give you exactly how that value $R$ (or distance to signal) effects the performance of the algorithm through our dynamics equation. Intuitively, the distance that you start from the "optimum" (that is denoted by essentially $R + \tilde{R}$) should mean that the algorithm takes longer to converge, which it does!

---

> > > > ### Comment · Reviewer_hZQR · 2022-08-07
> > > > **Follow-up discussion with the authors**
> > > >
> > > > Thanks again for the clarifications!
> > > >
> > > > - Regarding the limitation of $b = A \tilde x + \eta$, I think it should be emphasized that the setting in this paper is a _linear model_, in the introduction or abstract. In the latest response, the authors mentioned that 'as mentioned below, we can eliminate this assumption', but I don't see where this is explained. Did I miss anything?
> > > >
> > > > - Yes, I totally agree that the regime where $n$ is growing proportionally to $d$ is preferable.
> > > > - I get that due to the stochastic noise, the SGD+M iterates will oscillate around $\tilde x$ even if it's initialized exactly at $\tilde x$. This resolves my previous question.
> > > > - Regarding $R$ in Assumption 1, do you mean that $R$ can depend on $d$ and $n$, instead of being a universal constant? If so, then I have misunderstood this previously and I think this should be clarified in the paper because in Theorem 1, we are talking about the limit of large $n$ and $d$.

---

> > > > > ### Author Response · Authors · 2022-08-07
> > > > > **Comment**
> > > > >
> > > > > *First, we are happy to move the generative model assumption up further in the paper in order to make the paper more reader friendly; we will have to do this after the rebuttal, if accepted, because of the 9 page space limitation.*
> > > > >
> > > > > We thought the reviewer might also be interested in how to remove $R$ and $\tilde{R}$ or at least give a different interpretation especially in the setting when $d$ is fixed and $n \to \infty$.
> > > > >
> > > > > **1. Removing $R$ and $\tilde{R}$ and linear model**
> > > > >
> > > > > A way to remove $R$, $\tilde{R}$, and linear model (which we did not directly state in the paper, but it is hidden in the proof) is to view the whole forcing term $F(t) =  \frac{R}{2} h_1(t) + \frac{\tilde{R}}{2} h_0(t)$ as the loss function $f$ applied to full-batch gradient descent with momentum but with learning rate $\gamma \cdot \zeta$, $f(x_k^{\text{full batch gd+M}})$. We note the forcing term is the only place where $x_0, \tilde{x}$, and $\eta$ are used. They do not appear in the kernel.
> > > > >
> > > > > **Our assumptions on $x_0, \tilde{x},$ and $\eta$ involving the $R$ and $\tilde{R}$ only help simplify $f(x_k^{\text{full batch gd+M}})$, where $x_k^{\text{full batch gd+M}}$ is the $k$th iterate generated from full batch gradient descent with momentum and stepsize $\gamma \zeta$.** If you want to run simulations and have exact predictions without having to run GD+M, it is nice to have a simple form for $f(x_k^{\text{full batch gd+M}})$, but you could also leave it as is.
> > > > >
> > > > > You do need some mild assumptions on $x_0$ and $b$ (such as $\||x_0\||^2 \le C$ and $\||b\||^2 \le C$ independent of $n$ and $d$ plus a little more) to make everything work even in this case. For instance, you need this quantity $f(x_k^{\text{full batch gd+M}})$ to be defined for large $n$ and $d$.
> > > > >
> > > > > The dynamics would be
> > > > >
> > > > > $$ \psi(t+1) = f(x_k^{\text{full batch gd+M}}) + \gamma^2 \zeta (1-\zeta) \sum_{k=0}^t H_2(t-k) \psi(k), $$
> > > > > where $x_k^{\text{full batch gd+M}}$ is the $k$th iterate generated from full batch gradient descent with momentum and stepsize $\gamma \zeta$.
> > > > >
> > > > > This has the nice view of showing exactly that the loss under SGD+M is the loss under GD+M plus a noise term which is generated by the stochasticity in algorithm and we give explicit expressions for the noise.
> > > > >
> > > > > **2. High-dimensionality effect**
> > > > >
> > > > > We also wanted to point out the "high-dimensionality" effect is really not due to the linear model $b$. It's more subtle than this. The high-dimensionality allows us to simplify the noise generated from uniformly choosing indices of $A$ (or the noise generated by the stochastic algorithm). This is a nice reason why to assume high-dimensionality.
> > > > >
> > > > > **3. Interpretation of $R$ in Theorem 1**
> > > > >
> > > > > If you assume some mild assumptions like $||x_0||^2 \le C$ and $||b||^2 \le C$ independent of $n$ and $d$, Theorem 1 will change to the equation above in (1). The limiting loss value would become:
> > > > >
> > > > > $$\psi(\infty) = \frac{f(x^{\text{full batch gd+M}}_{\infty})}{1-||K||}$$
> > > > >
> > > > > where $f(x^{\text{full batch gd+M}}_{\infty})$ is the limiting value of the loss of full batch gradient descent with momentum but with stepsize $\gamma \zeta$. This quantity is well defined for any $x_0$ and $b$ set-up.
> > > > >
> > > > > *We hope the reviewer finds this an interesting avenue of research.*

---

> > > > > > ### Comment · Reviewer_hZQR · 2022-08-07
> > > > > > **Follow-up discussion with the authors**
> > > > > >
> > > > > > I would like to thank the authors for their efforts in explaining everything! Now the results make sense to me and I have adjusted the rating.
> > > > > >
> > > > > > Yes, please clarify the model assumption earlier in the paper when preparing future versions.
> > > > > >
> > > > > > The comments on the high-dimensionality effect seem interesting. Previously my impression was that it reflects the limiting spectral distribution of A as the law of A is assumed to be orthogonally (left) invariant. By 'simplify the noise generated from uniformly choosing indices of A', I guess what you are saying is related to the $A^\top P_k A$ part in Eq. (4)? I thought this form of update is crucial and somehow limited to the linear setting.

---

> > > > > > > ### Author Response · Authors · 2022-08-08
> > > > > > > **Follow-up from Reviewer hZQR (con't)**
> > > > > > >
> > > > > > > We really appreciate the reviewer's feedback and we will make the model assumptions appear earlier in the paper.
> > > > > > >
> > > > > > > **Related to the $A^TP_kA$**
> > > > > > >
> > > > > > > Yes, we were referring to the $A^TP_kA$ part in Eq (4). In order to get the dynamics with the Volterra equation, it roughly requires the variance of the mini-batch to also have a specific interpretation. For this, we require the data matrix $A$ to have some "delocalization of eigenvectors of AA^T", which many natural distributions of random matrices satisfy. We chose the strongest form of this delocalization, which is the left-orthogonal invariance condition. Some future work is to remove this condition to satisfy more random matrix ensembles.
> > > > > > >
> > > > > > > As the reviewer might know, delocalization of the eigenvectors does not directly imply anything about the spectral measure.  It is possible to create random matrices $A$ with arbitrary spectra (e.g., as non-concentrated as you want), but have left-orthogonal invariance.  But, it is true that eigenvector delocalization and concentrated spectral measure tend to ``happen at the same time'' for random matrices.
> > > > > > >
> > > > > > > ----
> > > > > > >
> > > > > > > Again, we thank the reviewer for the questions and the general interest in the results. We are happy to continue answering any questions the reviewer has.
> > > > > > >
> > > > > > > We also want to make aware to the reviewer that they have not yet clicked the *Author Rebuttal Acknowledgement* button on the top of the page. We want to make sure that the reviewer is acknowledged for their contributions to the author-reviewer discussion period.

---

> > > > > > > > ### Comment · Reviewer_hZQR · 2022-08-08
> > > > > > > > **Follow-up discussion with the authors**
> > > > > > > >
> > > > > > > > Thanks for the explanation! It makes perfect sense to me, and I can imagine that this enjoys certain degrees of universality.

---

### Official Review · Reviewer_n1XZ · 2022-07-11

**Rating:** 6
**Confidence:** 3
**Soundness:** 3 good
**Presentation:** 3 good
**Contribution:** 3 good

**Summary:**

The author analyzes the dynamics of large batch stochastic gradient descent with momentum and shows that the dynamics of SGD+M converge to a deterministic discrete Volterra equation as dimension increases. The author also identifies a stability measurement, the implicit conditioning ratio (ICR), for the stability of SGD+M and gives explicit choices for the learning rate and momentum parameter.

**Questions:**

1. When do we say the number of samples (n) is large? n = 1000?
2. Why does the author add $P_k$ in Eq. (4)?
3. Since the author uses the function $\psi$ to approximate the training loss trajectory, how do you select the factors, $R$ and $\tilde{R}$ such that $\psi$ can approximate the training loss trajectory accurately?

Typos

Line 86 used a -> used

Line 91 do -> does

Line 191 discussed -> are discussed

Line 211 a produce?

Line 303 benefit -> a benefit

Line 306 achieved -> be achieved

etc.


**Limitations:**

Yes

**Strengths And Weaknesses:**

Strengths:
1. The author provides a non-asymptotic comparison for the behavior of the training loss under SGD+M to a deterministic function $\psi$.
2. The exact loss trajectory gives a rigorous definition of the large batch and small batch regimes.
3. The author introduces a stability measurement that exactly captures the transition of SGD+M to an accelerated method.
4. The author finds that the dynamics of SGD+M and of SGD are truly non-equivalent.

Weaknesses
1. When the number of samples (n) is small, the function $\psi$ cannot accurately approximate the training loss.
2. Factors $R$ and $\tilde{R}$, should be predefined.

---

> ### Author Response · Authors · 2022-08-01
> **Response to Reviewer n1XZ**
>
> We thank the reviewer for their careful reading of our paper and thoughtful questions. We appreciate that the reviewer identified many contributions of this work. We fixed the typos and added in Appendix D an explanation of our hyperparameter tuning $R$ and $\tilde{R}$ in the *newly uploaded version of the paper and Supplementary Materials*.
>
> **Response to Weaknesses**
>
> *1. When the number of samples (n) is small, the function ψ cannot accurately approximate the training loss.*
>
> Yes, your discussion is exactly correct. When the problem size is small (say d = 30, n = 20), we have nothing to say. On the other hand, we would argue there is quite a bit of interest in large parameter and large sample settings with respect to the machine learning community.
>
> *2. Factors $R$ and $\tilde{R}$, should be predefined.*
>
> The value $\frac{R}{\tilde{R}}$ represents the signal-to-noise and indeed they can not readily be computed. However we note that the forcing term in Eq. 10, $F(t) = \frac{R}{2} h_1(t) + \frac{\tilde{R}}{2} h_0(t)$ has another interpretation as it equals the loss function $f$ under full-batch gradient descent with momentum with step size $\gamma \cdot \zeta$ (scaled by the batch fraction). Since the loss under full-batch gradient descent with momentum on a least squares problem is completely determined by the spectrum of the Hessian and the targets $b$ one does not actually need to compute the values of $\tilde{R}$ and $R$ [1].
>
> [1] C. Paquette, et al, Halting Time is Predictable for Large Models: A Universality Property and Average-Case Analysis, Found. of Computational Mathematics, 2022
>
> **Response to Questions**
>
> *1. When do we say the number of samples (n) is large? n = 1000?*
>
> In practice, we get good agreement with our theory when $n$ is in the hundreds. We note that the results hold for any $n$ and $d$. The statement of Theorem 1 says that as $n \to \infty$ the loss of mini-batch momentum concentrates around the deterministic function $\psi(t)$, meaning as the number of samples $n$ grows, the better the approximation of $\psi$ gets to the actual SGD+M.
>
> *2. Why does the author add $P_k$ in Eq. (4)?*
>
> The $P_k$ is a random projection matrix and its introduction was for notational convenience. Also it is convenient in the proof to think of the mini-batch as a random projection so that one can use results from high-dimensional probability. Note the $P_k$  is equivalent to randomly selecting rows of $A$ (or indices $i \in [n]$) as one would do in a mini batch stochastic algorithm.
>
> *3. Since the author uses the function $\psi$ to approximate the training loss trajectory, how do you select the factors, $R$ and $\tilde{R}$ such that $\psi$ can approximate the training loss trajectory accurately?*
>
> With generated data, such as $x_0 \sim N(0,I_d)$, $\tilde{x} \sim N(0, I_d)$, and $\eta \sim N(0, I_n)$, one can compute exactly $R$ and $\tilde{R}$. This was done to show proof of concept in Figure 1, for instance. On real data (e.g., Fig. 3), we grid searched through various values of $R$ and $\tilde{R}$. We added an explanation in Appendix D: Numerical Simulations explaining our hyperparameter tuning of $R$ and $\tilde{R}$. Alternatively, see Weakness point 2 above, $\frac{R}{2} h_1(t) + \frac{\tilde{R}}{2} h_0(t)$ has another interpretation as it equals the loss function $f$ under full-batch gradient descent with momentum with step size $\gamma \cdot \zeta$. One could run full-batch gradient descent with momentum (which is not stochastic) and get an expression for the $\frac{R}{2} h_1(t) + \frac{\tilde{R}}{2} h_0(t)$.

---

> > ### Comment · Reviewer_n1XZ · 2022-08-09
> > **Response to authors**
> >
> > Thanks for addressing questions in my original review and thanks for the clarification.

---

### Author Response · Authors · 2022-08-02
**Response to all reviewers**

We thank the reviewers for their comments and  suggestions. We encourage the reviewers to look at the **newly uploaded main paper and Supplementary Materials as well as the new experiments** of Figure 4 on MNIST data and convergence behavior near the ICR on the MNIST data set and simulated data **([see here for ICR near crossover regime](https://anonymous.4open.science/r/predicted_ICR-C2F8/figure7.pdf) and [here for Figure 4 on MNIST data](https://anonymous.4open.science/r/heat_maps-B391/figure6.pdf))**. In the new version of the paper, we have implemented suggestions by the reviewers including

1. Formatting changes (e.g., removed the figure wrapping and fixed typos)
2. Adding the definitions of $h_0, h_1$, and $H_2$ in Theorem 1
3. Rewriting parts of Section 2 to add more clarifications.
4. Adding new experiments showing parameter selection and convergence rate on real data. Particularly we redid Figure 4 with the MNIST data set (**[see Figure 6 in newly uploaded Supplementary Materials or here](https://anonymous.4open.science/r/heat_maps-B391/figure6.pdf)**)
5. Adding new experiments showing convergence rate near the ICR for simulated data and for MNIST data (**[see Figure 7 in newly uploaded Supplementary Materials or here](https://anonymous.4open.science/r/predicted_ICR-C2F8/figure7.pdf)**). The ICR exactly predicted the transition in convergence rates.

---

### Meta-Review · Area_Chair_Z9jC · 2022-08-26

**Recommendation:** Accept
**Confidence:** Certain

**Metareview:**

The paper analyses an SGD with Momentum (SGD+M) in a setting where the dimension and number of samples are large. The authors provide a theoretical justification for a least square problem.
They identify two settings based on the implicit conditioning ratio (ICR). In one setting, the SGD+M achieves linear convergence, whereas, for smaller batch sizes, the convergence speed gets worse.

In general, the paper presents novel ideas that provide new insides to a well-known and heavily used algorithm such as SGD+M. For a camera-ready version, please try to incorporate the valuable suggestions from reviewers. Thanks




**Award:**

No

---

### Decision · Program_Chairs · 2022-09-14

Accept